# The Brightest Light in Canada: The Canadian Light Source

**Jeffrey Cutler \*, Dean Chapman, Les Dallin and Robert Lamb**

Canadian Light Source Inc., University of Saskatchewan, Saskatoon, SK S7N 2V3, Canada;
Dean.Chapman@lightsource.ca (D.C.); Les.Dallin@lightsource.ca (L.D.); Robert.Lamb@lightsource.ca (R.L.)
\* Correspondence: jeffrey.cutler@lightsource.ca; Tel.: +1-306-657-3500

**Abstract:** Over forty years in the making, and one of Canada's largest scientific investments in those four decades, the Canadian Light Source (CLS), a third generation source of synchrotron light, was designed for high performance and flexibility and serves the diverse needs of the Canadian research community by providing brilliant light for applied and basic research programmes ranging from the far infrared to the hard X-ray regimes. Development of the scientific program at the CLS has been envisioned in four distinct phases. The first phase consists of the accelerator complex together with seven experimental facilities; the second phase adds six more experimental facilities and additional infrastructure to support them; the third phase adds seven more experimental facilities; and the fourth phase focuses on beamline and endstation upgrades, keeping the CLS as a state-of-the-art research centre. With the growth of a strong user community, the success of these experimental facilities will drive the future growth of the CLS.

**Keywords:** synchrotron light; beamlines; X-rays; infrared; spectroscopy; imaging

## 1. Introduction

The Canadian Light Source (CLS) is one of the largest scientific projects in Canada, and is classified as a third generation synchrotron light source. With annual operations of over 5000 h and thousands of users per year, the operating parameters of the CLS are very similar to those of most of the synchrotrons currently operational, making the CLS globally competitive. The electron beam energy is 2.9 GeV at an electron current in the accelerator of about 250 mA and a lifetime of about 20 h. When it was activated in 2005, it joined the ranks of an elite group of facilities. At the time, it was known as a pioneer in advanced light source and beamline design and operations with many of the CLS features being used in more recently commissioned facilities. The design of the CLS is the most compact of the newer machines, making it one of the most cost-effective to build and with a smaller facility footprint allowing for better integrations between operational divisions and the University of Saskatchewan, leading to a more cost-effective operation.

Every facility strives to have experimental infrastructure that is globally unique, setting the standard for advanced, state-of-the-art instrumentation and therefore enabling new and better science. The CLS has achieved that distinguished goal. All seven of its original beamlines are operating at an international level, and several are setting new standards in experimental parameters such as the scanning transmission X-ray microscopy (STXM) equipped with many different experimental environments including humidity cells, in situ catalysis, and cryo-tomography operating at spot sizes of 30 nm. The second phase of seven beamlines are operational and contain several beamlines that are truly unique for life [1] and materials sciences, as well as allow for new approaches for industrial utilization [2,3]. It is the third phase of three sectors with seven end stations under construction that will set a new international standard in both material characterization, a novel material understanding

for the next generation of quantum materials, and life sciences that is directly applicable to modern health. CLS beamlines cover a broad spectrum of research sectors and have been motivated and designed by outstanding world class scientists from across Canada, and will provide Canada with cutting-edge research leading to innovation.

## 2. Creation of the Canadian Light Source

The Canadian Light Source was born out of the growth of scientific expertise by Canadian researchers using offshore facilities—in particular, the highly successful Canadian Synchrotron Radiation Facility at the Synchrotron Radiation Center at the University of Wisconsin [4]. This success in the academic community and growing awareness in the industrial research community, coupled with the growing global recognition in all modern, industrially competitive nations of the importance of synchrotron radiation facilities, has led to the decision to fund a synchrotron in Canada. Once announced, a national competition took place to decide the location of the national facility. Based on the availability of funding at the provincial, city, and university level, and the availability of trained personnel and infrastructure from the Saskatchewan Accelerator Laboratory to support the development and operation of a synchrotron, the decision was made to build the CLS at the University of Saskatchewan. The formal funding announcement took place on 31 March 1999 and initial user operations began in 2005. Following 10 successful years of user operations and into the future, the CLS has a continuing vision to deliver innovative solutions as a leading centre for research excellence in health, agriculture, the environment, and advanced materials.

## 3. Experimental Facilities

The Canadian Light Source consists of a 300 MeV linear accelerator, a booster synchrotron to ramp the electron energy to 2.9 GeV, a storage ring with a very compact lattice [5] leading to exceptionally flexible performance that is coupled with a series of state-of-the-art experimental facilities. The accelerator complex has been reliable and flexible, easily supporting the needs of our users from single bunch mode for time resolved experiments to routine stable full energy operations. At present, the ring operates routinely at 250 mA of stored electron current with radio frequency power up-grade plans to move to a 500 mA operation and eventually moving to top-up operations. At present, the Canadian Light Source operates in a fill-decay mode where photon flux deteriorates over a 12 h fill cycle leading to temperature changes in optics and a decrease in beamline throughput due to lost time waiting for instrumentation to stabilize. It is anticipated that the CLS will transition to top-up operation in 2018 in which the beam current decay is minimized by frequent injections of electrons into the storage ring, allowing for an increased beamline operation and efficiency by having a constant heat load on optical components, thereby minimizing photon energy shifts, beam position drift, and increased instrumentation stability.

The CLS produces brilliant synchrotron radiation over an almost continuous range of wavelengths covering eight orders of magnitude. Delivery of reliable, stable, and intense fluxes of photons over this entire range is a testimony to the design of both the accelerator and experimental facility complexes.

Experimental facilities at the Canadian Light Source have been developed in a phased approach [6], and details can be found on the CLS website [7]. A plane view of these facilities can be seen in Figure 1, and a graphical representation of the available photon energies for each of these facilities is shown in Figure 2. It is apparent that, as the CLS matures, the space on the experimental floor is rapidly disappearing. With the completion of the third phase and planned beamline upgrades, only one straight section (the almost vertically pointing line at the top left of Figure 1) and seven bending magnet ports will still be available.

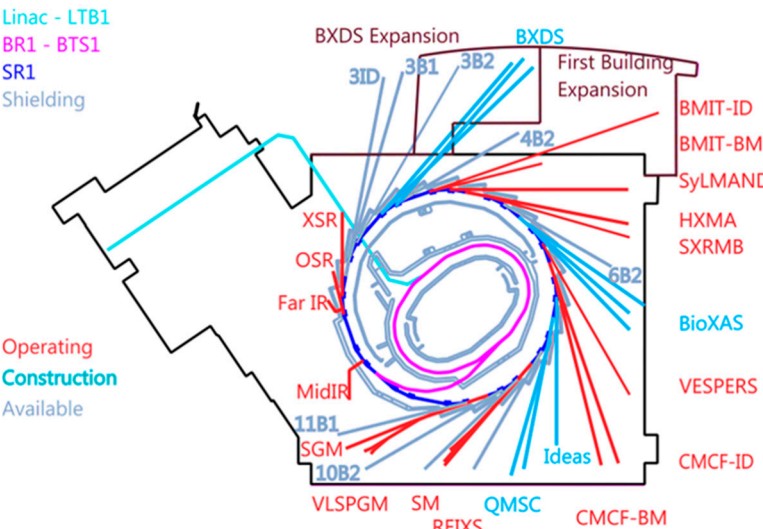

**Figure 1.** Schematic layout of the experimental hall at the Canadian Light Source (CLS).

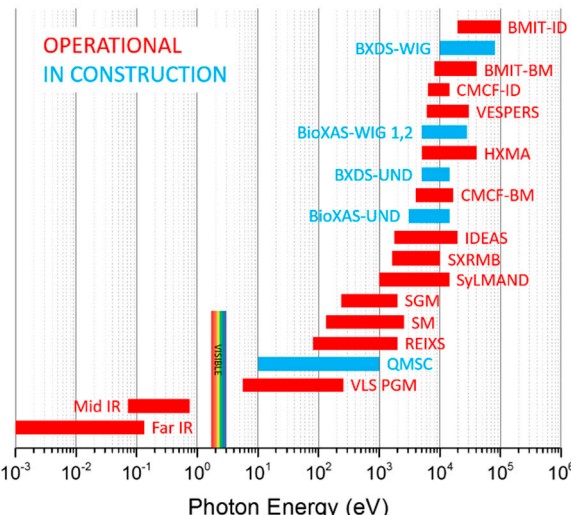

**Figure 2.** Available photon energy ranges for the operational and under construction experimental facilities. IR: Infrared.

## 3.1. Phase-I Experimental Facilities

The original funding for the infrastructure of $141 million, with an additional capitalization of $30 million of the existing Saskatchewan Accelerator Laboratory, was contributed by 18 organizations from governments, industries, and universities. With the funding from the Canada Foundation for Innovation (CFI), matching funding was committed from stakeholders from across Canada at all levels of government to make this a unique truly national facility. Stakeholders included Western Economic Diversification, the governments of Alberta, Ontario, and Saskatchewan, the City of Saskatoon, the Universities of Alberta, Saskatchewan, and Western, the Alberta Heritage Foundation for Medical Research, the National Research Council, Natural Resources Canada, Boehringer Ingelheim, and SaskPower.

The initial Phase-I experimental facility construction project has been completed, and the set of seven experimental stations and two diagnostic beamlines are operational. Scientific productivity continues to increase as the facilities achieve their ultimate design goals. The first phase was chosen to deliver the most essential "core" capabilities required by the majority of researchers who were part of

the "original" user community. It was part of the initial development of the CLS facility that extensive interaction with the research community through a coast-to-coast lecture and workshop campaign identified facilities that most urgently needed to deliver on our mandate of excellence in basic and applied research.

This first phase development supported a rich variety of user programs (in addition to the X-ray synchrotron radiation (XSR) and optical synchrotron radiation (OSR) diagnostic facilities): high-resolution far infrared spectroscopy (far IR), mid-infrared spectromicroscopy (mid-IR), a variable line spacing plane grating monochromator (VLS-PGM) beamline, a high-resolution spherical grating monochromator (SGM) beamline, soft X-ray spectromicroscopy (SM), hard X-ray micro-analysis (HXMA), and the Canadian Macromolecular Crystallography Facility (CMCF ID), with each of these Phase-I facilities supporting an active and growing user community and generating high quality data suitable for scientific publication.

The far IR experimental facility (02B1-1) uses a bending magnet source and point-to-point optics to transfer light from the storage ring to a high-resolution Bruker IFS125HR infrared spectrometer (Ettlingen, Germany) with a 9-compartment scanning arm, producing a resolution of better than 0.001 wavenumbers in the spectral range from 5 to 1000 cm$^{-1}$ with a flux of $1 \times 10^{13}$ $\gamma$/s/0.1% bandwidth (BW) at 1000 cm$^{-1}$. This facility is optimized for the analysis of a gas phase species of atmospherical and astrophysical interest.

Using a similar optical scheme as that of the far IR beamline, the mid-IR spectromicroscopy facility (01B1-1) provides diffraction limited spatial resolution (between 4000 cm$^{-1}$ and 400 cm$^{-1}$) through the combination of a Bruker Vertex 70v infrared spectrometer and Hyperion 3000 confocal microscope/mapping stage.

The VLS-PGM beamline (11ID-2) uses a 185 mm planar undulator as its source, and delivers monochromatic 5.5–250 eV light with a flux of $2 \times 10^{11}$ $\gamma$/s/0.1% BW into a 500 μm × 500 μm spot and a resolving power greater than 10,000 into one of two parallel endstations for X-ray absorption spectroscopy (XAS), X-ray photoelectron spectroscopy (XPS), and X-ray excited optical luminescence (XEOL) spectroscopy. These endstations can be used for a broad cross section of research interests ranging from environmental to advanced materials. For example, these techniques can be readily applied to better our understanding for a clean technology future, including the advancement of battery technology ideal for electric vehicles. By applying the surface sensitivity of XPS, the byproducts produced by sodium-air batteries could be characterized, allowing for improvements in future manufacturing and utilization [8].

The SGM beamline (11ID-1) uses a 45 mm planar undulator as its source, and provides monochromatic photons in the 240–2000 eV spectral range with a flux of $4 \times 10^{12}$ $\gamma$/s/0.1% BW at 250 eV into a 1000 μm × 100 μm spot and a resolving power greater than 5000. Separate toroidal focusing mirrors deliver light to one of two endstation locations arranged in a pass-through geometry for XAS, XPS, XEOL spectroscopy, gas phase photoionization, and Time of flight (TOF) measurements.

Soft X-ray spectromicroscopy (10ID-1) has an Apple II-type undulator delivering photons of arbitrary polarization in the spectral region from 130 to 2700 eV using a plane grating monochromator that provides a flux of $10^8$ $\gamma$/s/0.1% BW in a 30 nm spot with a E/ΔE of 3000–7000 resolving power. The optical design of this facility supports one of two parallel endstations—scanning transmission X-ray microscopy (STXM) and a photoelectron emission microscope (PEEM), which provides new insight on challenging microscopy questions. When coupled with other techniques, such as mid-IR microscopy, significant insights can be obtained about the world around us such as in the field of crop utilization. By applying both techniques, agricultural researchers were able to identify and map the distribution of complex biopolymers, which could ultimately lead to improved plant quality and new bioproduct development [9].

The Hard X-ray MicroAnalysis (HXMA) facility (06ID-1) uses a superconducting wiggler (optimized for brilliance by having a relatively low K) to support a number of experimental programs, including micro- and bulk X-ray absorption fine structure (XAFS) spectroscopy and X-ray diffraction

(XRD). A careful design of the optics allows easy switching between these different modes and accommodates photons from 5000 to 50,000 eV with a flux of $10^{12}$ γ/s/0.1% BW at 12,000 eV and a resolving power of 10,000 into an 800 μm × 1500 μm spot.

HXMA can play a role in many research fields ranging from mining to advanced materials. For example, in the field of clean technology, soft (SGM) and hard X-ray techniques (HXMA) can bring new insight to important modern energy questions including the production of hydrogen from water splitting for fuel cell operations by better understanding more efficient catalysts that can be produced from abundant low-cost metals [10].

The CMCF ID facility (08ID-1) uses a 20 mm hybrid small gap in-vacuum undulator to provide brilliant light from 6200 to 18,000 eV with a flux of 2 × $10^{12}$ γ/s/0.1% BW at 12,000 eV and a resolving power of 10,000 into a 130 μm × 30 μm spot. This enables protein crystallographic studies of small crystals and crystals with large unit cells using single crystal X-ray diffraction, multiwavelength anomalous dispersion (MAD), and XAFS on crystals. For instance, in the area of health, with an ability to examine smaller protein crystals, a team from the University of British Columbia was able to gain insight into the development of a universal blood type by developing an enzyme that removes A- and B-type antigens from blood, which may improve the availability of life-saving procedures by limiting the need for matching blood types [11].

### 3.2. Phase-II Experimental Facilities

The second phase facilities at the CLS were developed through a second extensive consultation with our user community. A number of exciting new capabilities complement the first phase facilities and extend them to support our planned research sectors of pre-eminence: agriculture, health, environment, and advanced materials. In the Phase-II competition, seven additional beamlines were added at a total cost (CFI plus matching funds from 3 provinces and industry) of $55 million with matching funds from the Provinces of Saskatchewan, Ontario, and British Columbia.

The Phase-II facilities include resonant elastic and inelastic X-ray scattering (REIXS), a soft X-ray microcharacterization beamline (SXRMB), the Synchrotron Laboratory for Micro- and Nano-Devices (SyLMAND), the High Throughput Canadian Macromolecular Crystallography Facility (CMCF BM), Very Sensitive Elemental and Structural Probe Employing Radiation from a Synchrotron (VESPERS), the Industry, Development, Education, and Students (IDEAS) beamline, and two facilities, both a bending magnet and insertion device, for BioMedical Imaging and Therapy (BMIT).

REIXS (10ID-2) uses an Apple II-type elliptically polarizing undulator to provide 80–2000 eV photons with a flux of $10^{12}$ γ/s/0.1% BW at 100 eV into a 250 μm × 150 μm spot and with arbitrary polarization to one of two experimental endstations (X-ray emission spectroscopy and resonant soft X-ray scattering). This soft X-ray scattering facility combined with in situ sample preparation allows for the study of a wide range of advanced and novel materials. REIXS and SM will have similar insertion devices, and part of their operation will involve "rapid polarization switching" enabled by turning off one beamline and illuminating the other with both insertion devices, selected by a chopper. With novel materials come novel insights, for example in the field of high-temperature superconductors, where a better understanding of how charge density wave instability competes with superconductivity could aid in bringing superconductors into large-scale real world applications [12].

SXRMB (06B1-1) is a medium energy beamline that uses a bending magnet source and covers a photon energy range of 1700 to 10,000 eV, using InSb(111) and Si(111) crystals with a flux of $10^{11}$ γ/s/0.1% BW into a 300 μm × 300 μm spot. It is equipped with four major endstations, namely two bulk X-ray absorption setups (under vacuum and ambient conditions under helium), which can be used for dynamic catalyst studies, a hard X-ray photoemission spectrometer, and a microprobe endstation based on Kirkpatrick–Baez focusing with a nominal spot size of 10 μm × 10 μm.

SXRMB has been used to answer a number of fundamental and applied agriculture and food security questions including increasing our understanding of Saskatchewan soil which are among the most nutrient-rich in the world. Researchers from the University of Saskatchewan studied the

behaviour of phosphorous in various soil and fertilizers types and landscapes in order to better understand the mechanism of phosphorous retention that will help improve our strategies for fertilizer application [13].

SyLMAND (05B2-1) uses a bending magnet source, a double mirror energy filter, and a sophisticated sample scanner to provide 1000–15,000 eV light with a beam power of 700 W in a 15 mm (vertical) by 150 mm (horizontal) fan for deep X-ray lithography. This facility allows the fabrication of a wide variety of micro-electro-mechanical systems. The facility also includes a well-equipped process clean room.

CMCF BM (08B1-1) provides 4000–18,000 eV light from a bending magnet source with a flux of $10^{11}$ γ/s/0.1% BW at 12,000 eV into a 230 μm × 195 μm spot to an automated robotic sample handling facility optimized for high throughput multi-wavelength anomalous dispersion protein crystallography. Moving beyond its base applications, CMCF BM has found new communities by supporting a growing service work in the areas of powder X-ray diffraction and small molecule single crystal diffraction.

VESPERS (07B2-1) provides 6000–30,000 eV photons with a flux of $10^9$ γ/s/0.1% BW at 15,000 eV into a spot of 3 μm × 3 μm and for simultaneous X-ray diffraction and X-ray fluorescence spectroscopic analysis of micro-volumes of sample. Multi-bandpass and pink beam capabilities are provided, and the optics are designed for rapid switching between these applications. The beamline control system is designed to enable remote operation of the facility. With an ability to image metal distribution on the micron size scale, scientists from the University of Saskatchewan were able to provide new insight into the fate of crew members of the HMS Terror of the Franklin mission that disappeared looking for the Northwest Passage in the mid-1800s. By examining nail tissue, the study identified a zinc deficiency related to malnutrition, which played a role in the lead poisoning of the crew [14].

BMIT is a research facility designed to image biological tissues in small to large animals and to do research in support of therapeutic applications of X-rays in living organisms. The bending magnet facility (05B1-1) will provide 15,000–40,000 eV photons with a flux of $1.5 \times 10^{11}$ γ/s/0.1% BW at 10,000 eV and a beam size of 240 mm × 7 mm and the spectral range for the insertion device (05ID-2), a novel superconducting wiggler, is from 20,000 to over 100,000 eV with a flux of $3 \times 10^{12}$ γ/s/0.1% BW at 20,000 eV and a beam size of 220 mm × 11 mm. This innovative facility supports and develops a large range of X-ray based imaging and therapeutic modalities. For example, the BMIT insertion device beamline was used to peer inside blown-up batteries by applying computed tomography to examine the structural changes that occur inside a lithium ion battery following overcharging, leading to gas generation that is trapped inside a battery pack and that further leads to pillowing, which can decrease performance or cause the battery to leak [15].

IDEAS (08B2-1) is a multipurpose bending magnet beamline providing 3400–13,500 eV photons for long-term research studies, technique research, and development along with the training of students and new users. The robust beamline can accommodate a broad cross section of endstations with an initial focus on in situ X-ray absorption and X-ray fluorescence studies.

### 3.3. Phase-III Experimental Facilities

As with the other development phases, the third phase of our development followed a series of workshops to identify community requirements. As a result of these workshops, three new insertion device-based facilities were recommended to augment CLS capabilities. An additional seven beamlines were awarded in Phase-III. The total capital investment by CFI, provinces, and industry in Phase-III is $65 million with matching funds from the Provinces of Saskatchewan, Ontario, British Columbia, and Quebec. The new facilities most directly affect our life and material sciences programs. When this third phase is completed, the CLS will be a mature facility capable of supporting most of the synchrotron-based research programmes envisioned by our community. The emphasis is on providing flexible state-of-the-art facilities in each developed area.

In the field of material science, the Quantum Materials Spectroscopy Centre (QMSC) (09ID-1) provides arbitrary polarization photons from one of two parallel elliptically polarizing undulators (225 and 55 mm periods), covering an energy range of 10–1200 eV with a flux of $10^{13}$ $\gamma$/s/0.1% BW into a 20 $\mu$m $\times$ 100 $\mu$m spot to probe the low energy electronic properties of novel materials using spin and angle resolved photoemission spectroscopy. Separate organic and inorganic molecular beam epitaxy chambers, and a ultra-high vacuum (UHV) distribution chamber, will enable this material science programme.

Brockhouse X-ray Diffraction and Scattering Sector (BXDS) (04ID-1) will support research programmes in life, environmental, and material sciences by providing photons in the 3000–80,000 eV range. The facility consists of two separate beamlines; one has a high brilliance cryogenic in-vacuum undulator (3000–20,000 eV) with a flux of $10^{12}$ $\gamma$/s/0.1% BW into a 400 $\mu$m $\times$ 50 $\mu$m spot, and the other a high flux superconducting wiggler (30,000–100,000 eV) with a flux of $10^{12}$ $\gamma$/s/0.1% BW into a 2000 $\mu$m $\times$ 20 $\mu$m spot at 80,000 eV. While the beamlines share a straight section, they operate independently of each other. In addition to resonant and non-resonant hard X-ray scattering, the facility will also enable small and wide angle X-ray scattering.

The life science beamline for X-ray absorption spectroscopy (BioXAS) (07ID-1) is specifically designed to support life and environmental science research using X-ray absorption spectroscopy and XAS imaging on a wide range of length scales "from the atomic to the anatomic" with ultra-low sensitivity and very high resolution. For example, it will enable the analysis of metals both in biological molecules and in whole tissues. The facility consists of three independently operated beamlines with two dedicated for X-ray absorption spectroscopy, which share a permanent magnet wiggler (5000–28,000 eV) with a flux of 9 $\times$ $10^{12}$ $\gamma$/s/0.1% BW into a 600 $\mu$m $\times$ 200 $\mu$m spot. The third beamline is equipped with an in-vacuum undulator (5000–21,000 eV) with a flux of 2 $\times$ $10^{13}$ $\gamma$/s/0.1% BW into a 400 $\mu$m $\times$ 20 $\mu$m spot and designed for multi-mode fluorescence imaging.

### 3.4. Phase-IV Experimental Facilities

In order to remain state-of-the-art, many beamline facilities require significant upgrades of their core infrastructure. Reflecting their high user demand and changes in beamline technologies, our protein crystallography insertion beamline (CMCF ID) and two of our soft X-ray beamlines (VLS-PGM and SGM) are being upgraded. For CMCF ID, the upgrade includes a Pilatus pixelated detector, a faster robot, a new insertion device, and monochromator upgrades. As for the two soft X-ray beamlines, the upgrades include new Apple-II EPUs, microfocussing capabilities, and new endstations that will allow for more in situ and dynamic studies.

### 3.5. Partnership with the Advanced Photon Source

In order to support a broader synchrotron user community that expands beyond current capabilities and capacity, the CLS has partnered with the Advanced Photon Source located at Argonne National Laboratories (Lemont, Il, USA) to make a series of X-ray Science Division (XDS) beamlines and dedicated staff available to Canadian users. In particular, the Canadian users have made use of Sector 20 insertion and bending magnet beamline facilities that include confocal X-ray microscopy, X-ray Raman scattering, and resonant X-ray emission spectroscopy.

## 4. CLS Scientific Community

Fundamental to the operation of this key national facility is the principle of "excellence of science." In support of the diverse community that uses the CLS as part of their research programs, the CLS constructs and operates a wide diversity of instrumental tools, strives for excellence in science, and provides access to academic, government, and industrial scientists. A synchrotron is often characterized as a "Big Science" facility, when in fact it is a large source of electromagnetic radiation simultaneously serving a very large number of "Small Science" stations. The available facilities at the CLS cover all fields of the natural sciences that utilize infra-red, ultra-violet, or X-rays

to characterize the properties of matter, alive or inert. Major programs in surface science, health sciences, life-sciences, geology, and nano-technology, and manufacture of micro-devices, as well as many cross-linked interdisciplinary relations between these fields, are just some representative examples of the breadth of science and applications found at the Canadian Light Source.

The Canadian Light Source formally opened in 2005 as the culmination of a five-year construction project was complete. Since the first experimental facility was made available in 2004, the number of operational beamlines has increased to 15 with a growth in the number of individual users accessing the facility reaching over 1400 and by 2015 leading to thousands of publications in the peer-reviewed literature. As the number of beamlines increases to its presently funded level of 22, it is estimated that the annual number of individuals accessing the facility will be about 2000, amounting to 2500 user visits to the CLS including a significant number of international clients. As of December 2015, scientists from 10 provinces and 2 territories and 28 foreign countries have performed experiments at the CLS, and these numbers continue to grow.

## 5. Globally Competitive Industrial Program

The vision and mission of the Canadian Light Source is to serve both the academic and industrial communities in order to ensure that Canadian industry is globally competitive and that the synchrotron scientific output provides a better quality of life to all Canadians. CLS was the first synchrotron project ever to begin building an industrial and business development effort at the very inception of the project. The targets set for industrial participation, and resulting revenue, are one of the most aggressive at any synchrotron facility. The path to success is challenging, but CLSI is determined to be the world leader in synchrotron industrial utilization [2,3]. CLS is "breaking ground" by actively approaching commercial entities from various industries, examining their operational issues, and designing synchrotron-based solutions to address and practically solve these real business problems. To achieve this goal, CLS makes up to 25% of all allocated time for each beamline available for use by industry on a fee-access basis. The intellectual property policies are designed to be very attractive for a company desiring to use a synchrotron. In addition, CLS has a group of world-class scientists dedicated to assisting industrial partners. These efforts continue to grow as more and more resources become available.

## 6. Economic Impact

As a national scientific research resource, CLS is positioned to be one of Canada's leading partners in generating economic and social benefits through scientific discovery and applications, as well as fostering university/industry partnerships.

Economic impact can be measured in many ways, also over many time scales. A number of outcomes can be envisaged such as new businesses, improved products, training and retention of highly qualified personnel, increased investment in Canada in technology, business and people, direct impact on the economies at all levels of government, and the impact on society as a whole. The impact of these outcomes is difficult to quantify, but in simple terms CLS can point to the direct economic impact of its construction and development that utilized approximately 70% of Canadian contractors and suppliers. There are numerous examples of development of supplier expertise in design and development of scientific equipment. CLS has already provided a technology transfer of equipment components to Canadian businesses to enable the commercial exploitation of designs.

The research performed at the CLS will undoubtedly lead to new drugs and new medical and environmental methods, which will lead to new economic growth and enhancements to the quality of life of all Canadians. The fact that the CLS, from its inception, has had a very well defined and structured program related to business development and industrial science, and is certainly one of Canada's leading centres for training, attraction, and retention of highly trained personnel, assures that real benefits to Canadians are forthcoming.

## 7. Looking Forward

With the development of a forward looking strategic plan from 2017 to 2022, the CLS has identified several key objectives to support our broad user community while at the same time begin to plan for a new facility with objectives that include the following:

- increase impact by being a solution provider by cultivating high performance clients and increasing support to ensure the success of new clients;
- focus research and industry sector activities in areas with the greatest potential including agriculture, environment, health and advanced materials;
- enhance and capitalize on machine science and engineering expertise;
- establish a broad awareness of the CLS as an accessible driver of innovation.

In concert with a forward looking strategic plan, the CLS has commenced a planning process for a new low-emittance light source that will support our current clients while energizing a new community of CLS users who conduct experiments at facilities around the world. By the end of 2018, it is anticipated that the CLS will have developed a science case that requires access to a low-emittance light source along with the machine conceptual design that will keep Canadian science and innovation in a leadership position.

## 8. Conclusions and Speculations

Following the completion of Phase-IV beamline upgrades and as we look to the future, the CLS will have a suite of visualization tools whose combined range and capability will be almost unique on the international scale. Each of the facilities is independently competitive with the world standard and will allow the CLS to offer a research environment that will encourage the recruitment and retention of world-class scientists on the Canadian scene.

As with any maturing facility, the increasingly sophisticated and diverse research programmes developed by the user community and by our own scientists will drive the future.

**Acknowledgments:** The research described in this paper was performed at the Canadian Light Source, which is supported by the Canada Foundation for Innovation, Natural Sciences and Engineering Research Council of Canada, the University of Saskatchewan, the Government of Saskatchewan, Western Economic Diversification Canada, the National Research Council Canada, and the Canadian Institutes of Health Research. Sector 20 facilities at the Advanced Photon Source, and research at these facilities, are supported by the US Department of Energy—Basic Energy Sciences, the Canadian Light Source, and its funding partners, the University of Washington, and the Advanced Photon Source. Use of the Advanced Photon Source, an Office of Science User Facility operated for the U.S. Department of Energy (DOE) Office of Science by Argonne National Laboratory, was supported by the U.S. DOE under Contract No. DE-AC02-06CH11357.

**Author Contributions:** J.C., D.C., L.D., and R.L. conceived and wrote the paper.

**Conflicts of Interest:** The authors declare no conflict of interest. The founding sponsors had no role in the design of the study; in the collection, analyses, or interpretation of data; in the writing of the manuscript; or in the decision to publish the results.

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
