# Peer review of "The Brightest Light in Canada: The Canadian Light Source"

_qubs, doi:10.3390/qubs1010004_

Round 1

Reviewer 1 Report

The authors present a review of the Canadian Light Source after 11 years of operation. After shortly introducing the facility and its creation, the authors list the experimental stations that are available (phase I, phase II and phase III of the beamline construction program) and present shortly the phase IV upgrade. They then explain how industrial access is unique with respect to other synchrotrons, by providing a large fraction of the time to industrial users. This short review is well written and details the different beamlines. It should however been improved by substantiating some of the claims made in the introduction:

- Why is the facility the most cost effective to operate? 

-  Several beamlines are setting new standards in experimental parameters? It would be important to highlight in the text which ones are for which reasons. 

If these minor corrections are applied, the paper is fit for publication 

Author Response

The authors appreciate the reviewers comments which made the manuscript stronger.  The updated manuscript address the two questions asked by the reviewer.

Question 1: Why is the facility the most cost effective to operate?

The updated manuscript expands on this point in the last line of the first paragraph (line 33) and by highlighting the fact that with a smaller footprint allows for better integration between operational divisions as well as with the facility owner, the University of Saskatchewan, leading to improved efficiencies in our operations.

Question 2: Several beamlines are setting new standards in experimental parameters? It would be important to highlight in the text which ones are for which reasons.

The update manuscripts highlights and expands on how a number of beamlines are meeting or setting industry standards for experimental operations.  For example, in paragraph 2 (line 39), the manuscripts explains that various beamlines have a broad breadth of tools allowing for in situ and dynamic studies including SM which has humidity cell, in situ catalysis setup and cryo-tomograph,  Along with the expanded point in paragraph 2, the manuscripts now higlights a number of examples of cutting edge research that is on-going on many of the beamlines with impact on a broad array of research areas.

Reviewer 2 Report

Perhaps some major CLS references should be added to the Introduction.

Section 3 on Experimental Facilities might be made more interesting for the reader if examples of the science and/or research results were included for each of the described beam lines.  At the moment this section is pretty dry.

The typical metrics for light source performance - brightness and/or spectral flux density into an experiment acceptance aperture - are nowhere mentioned in the article.

Most light source accelerator facilities have an ongoing plans for enhancing performance but, other than going to 500 mA and top-up injection, these are not mentioned.  If there are any significant future performance enhancing plans, besides top-up and 500 mA, it might be good to mention them.

A short description of what top-up injection is and why it is beneficial would be useful the reader.

The color coding in Fig 1 might be improved --- there are 2 slightly different shades of red and 2 slightly different shades of teal that are not easily distinguishable.

Author Response

The authors appreciate the reviewers comments which made the manuscript stronger.  The updated manuscript address the six points raised by the reviewer.

Point 1: Perhaps some major CLS references should be added to the Introduction.

A number of new references have been added to manuscript to give a snapshot of the types and range of work being done at the Canadian Light Source.  On any given year, over 200 peer-review publications are generated by our user community.

Point 2: Section 3 on Experimental Facilities might be made more interesting for the reader if examples of the science and/or research results were included for each of the described beam lines.  At the moment this section is pretty dry.

A number of new references and highlights have been added to the manuscript to give a snapshot of the types and range of work being done at the Canadian Light Source.  On any given year, over 200 peer-review publications are generated by our user community.  In particular, most beamline descriptions now highlight an example of work done using the facilities at the CLS. For example, line 138 gives an example of the energy storage work on-going on the VLS-PGM beamline or line 153 hightlights the applications two beamlines (Mid-IR and SM) to address a single crop science based question.

Point3: The typical metrics for light source performance - brightness and/or spectral flux density into an experiment acceptance aperture - are nowhere mentioned in the article.

The typical fluxes and beam sizes have been added for all of the beamlines.

Point 4: Most light source accelerator facilities have an ongoing plan for enhancing performance but, other than going to 500 mA and top-up injection, these are not mentioned.  If there are any significant future performance enhancing plans, besides top-up and 500 mA, it might be good to mention them.

Lines 351 to 366 highlight the new CLS strategic plan and objectives for the years 2017 to 2022.  The section on looking forward discusses the CLS plan on developing a new low emittance light source.

Point 5: A short description of what top-up injection is and why it is beneficial would be useful the reader.

Lines 74 to 78 highlight the importance of top up to routine operations of the facility and how they can improve the stability and reproducibility of the beamlines.

Point 6: The color coding in Fig 1 might be improved --- there are 2 slightly different shades of red and 2 slightly different shades of teal that are not easily distinguishable.

Colours have been updated in order to be consistent between the Figures 1 and 2.